# A temporal knowledge graph reasoning model based on recurrent encoding and contrastive learning

Weitong Liu[1,2], Khairunnisa Hasikin[3], Anis Salwa Mohd Khairuddin[2], Meizhen Liu[1,2] and Xuechen Zhao[1]

[1] School of Data and Computer Science, Shandong Women's University, Shandong, China
[2] Department of Electrical Engineering, Faculty of Engineering, Universiti Malaya, Kuala Lumpur, Malaysia
[3] Department of Biomedical Engineering, Faculty of Engineering, Universiti Malaya, Kuala Lumpur, Malaysia



Corresponding author
Khairunnisa Hasikin,
khairunnisa@um.edu.my

## ABSTRACT

Temporal knowledge graphs (TKGs) are critical tools for capturing the dynamic nature of facts that evolve over time, making them highly valuable in a broad spectrum of intelligent applications. In the domain of temporal knowledge graph extrapolation reasoning, the prediction of future occurrences is of great significance and presents considerable obstacles. While current models consider the fact changes over time and recognize that historical facts may recur, they often overlook the influence of past events on future predictions. Motivated by these considerations, this work introduces a novel temporal knowledge graph reasoning model, named Temporal Reasoning with Recurrent Encoding and Contrastive Learning (TRCL), which integrates recurrent encoding and contrastive learning techniques. The proposed model has the ability to capture the evolution of historical facts, generating representations of entities and relationships through recurrent encoding. Additionally, TRCL incorporates a global historical matrix to account for repeated historical occurrences and employs contrastive learning to alleviate the interference of historical facts in predicting future events. The TKG reasoning outcomes are subsequently derived through a time decoder. A quantity of experiments conducted on four benchmark datasets demonstrate the exceptional performance of the proposed TRCL model across a range of metrics, surpassing state-of-the-art TKG reasoning models. When compared to the strong baseline Time-Guided Recurrent Graph Network (TiRGN) model, the proposed TRCL achieves 1.03% improvements on ICEWS14 using mean reciprocal rank (MRR) evaluation metric. This innovative proposed method not only enhances the accuracy of TKG extrapolation, but also sets a new standard for robustness in dynamic knowledge graph applications, paving the way for future research and practical applications in predictive intelligence systems.

## INTRODUCTION

Knowledge graphs (KGs) act as repositories of factual information within the human world comprising a large array of data utilized throughout different intelligent

applications. These applications extend from information retrieval (*Gaur et al., 2022*; *Liu et al., 2018*; *Wu et al., 2023a*), recommendation systems (*Wang et al., 2018*, *2019*; *Zhang et al., 2021*, *2023*), and question answering (*Bakhshi et al., 2022*; *Zhang et al., 2022b*, *2022c*). However, given the dynamic nature of facts, conventional KGs struggle to accurately depict their evolution. To address this limitation, the notion of temporal knowledge graphs (TKGs) has emerged. Temporal knowledge graphs are designed to capture the temporal aspect of facts enabling the representation of evolving information over time. Each event in TKGs can be expressed by a quadruple alongside a timestamp, *i.e.*, (subject entity, relation, object entity, timestamp). For example, a quadruple (NBA, champion, Golden State Warriors, 2017) represents the Golden State Warriors won the NBA champion in 2017.

Although existing TKGs encompass a wealth of information, they often encounter issue with incomplete data. This has led to the increasing trends of research in TKG reasoning. Thus, TKG reasoning aims to fill these gaps by making predictions on missing entities. For example, the TKG model is based on historical facts about NBA champions before 2018 to complete query (NBA, champion, ?, 2018). TKG reasoning can be separated into two types according to different reasoning timestamps, namely interpolation reasoning and extrapolation reasoning (*Jin et al., 2019*). Given the temporal knowledge graph with timestamps ranging from t1 to tn, the interpolation reasoning (*Wu et al., 2022*; *Xu et al., 2020*) objective is to infer lost facts that occurred at timestamp t, where $t_1 < t < t_n$. Unlike the interpolation reasoning objective, the extrapolation reasoning (*Li et al., 2021b*; *Zhu et al., 2021*) objective is to forecast facts that occur at future timestamp t, where $t < t_n$. Therefore, the extrapolation reasoning objective can predict events that happen in the future, which is much more practical and challenging. Our work is to study TKG reasoning based on extrapolation reasoning.

At present, a prevailing approach in temporal knowledge graphs extrapolation reasoning involves structuring temporal knowledge graphs into graph representations. This group entails representing the fact set corresponding to each timestamp as a snapshot within a graph structure. Then, the graph convolution network (GCN) model is employed to identify the graph structure representation of the KG snapshot corresponding to each timestamp, while a double recurrent mechanism is utilized to seize temporal interactions among diverse knowledge graph (KG) snapshots. This approach comprehensively accounts for the evolution of historical facts. Elucidating dependency relationships among them. Prominent models, such as recurrent event network (RE-NET) (*Jin et al., 2019*), GCN-based Recurrent Evolution network (RE-GCN) (*Li et al., 2021b*), and Complex Evolutional Network (CEN) (*Li et al., 2022a*) models exemplify this approach. Some workers have also explored the influence of recurring historical data on TKG reasoning, as evidenced by Temporal Copy-Generation Network (CyGNet) (*Zhu et al., 2021*) and Contrastive Event Network (CENET) (*Xu et al., 2023b*) models. The CENET model, incorporates contrastive learning aim to learn the correlation between historical and non-historical facts, albeit overlooking the evolving nature of historical facts resulting in unsatisfactory experimental outcomes. Meanwhile, the TiRGN (*Li, Sun & Zhao, 2022*)

model uses the GCN and the gated recurrent unit to obtain the dependency relationship of historical facts, and this model also considers the impact of repeated history facts on future entity prediction. However, this model ignores the potential interference of repeated history facts on future fact prediction. According to the aforesaid issues, this work put forward a original TKG reasoning model, termed Temporal Reasoning with Recurrent Encoding and Contrastive Learning (TRCL). This model employs recurrent encoding to capture the dynamic relationships among facts, generating representations of entities and relationships. Then, the model integrates a global historical matrix to account for the influence of repeated historical facts on entity prediction. In addition, leveraging contrastive learning, TRCL mitigates the interference of historical facts on future entity prediction. Finally, the model obtains the results of TKG inference tasks through a time decoder. Therefore, the primary contributions of this article are as described below:

- A TRCL model for TKG reasoning is proposed, which not only captures the dependency relationship among historical facts but also addresses the positive and negative influences of repeated historical facts on entity prediction.

- A recurrent encoder is developed using the graph convolution network and the double recurrent mechanism. This encoder effectively captures the dependency relationships among historical facts. Additionally, this work proposes a historical matrix that comprehensively accounts for repeated historical facts. In addition, the TRCL model integrates contrastive learning to alleviate the impact of irrelevant historical data on entity prediction. Finally, the developed model uses a decoder containing periodic time vectors to derive TKG reasoning outcomes.

- Extensive testing was performed on multiple public datasets to strengthen the validity and generalizability of the TRCL model providing robust evidence on its effectiveness across diverse scenarios.

## RELATED WORK

### Temporal knowledge graph reasoning

TKG reasoning is segmented into interpolation reasoning and extrapolation reasoning based on different timestamps. Interpolation reasoning refers to inferring facts that are missing in historical timestamp. For example, Temporal TransE (TTransE) (*Leblay & Chekol, 2018*) incorporates timestamp vectors to represent temporal transformations between entities. Hyperplane-based Temporally aware knowledge graph Embedding (HyTE) introduced by *Dasgupta, Ray & Talukdar (2018)* represents time as a hyperplane projects entities and relationships onto it to encode temporal information. TNTComplEx (*Lacroix, Obozinski & Usunier, 2020*) employed tensor factorization, to model events with added temporal information as fourth-order tensors. However, these methods have poor ability to capture the evolution of facts and are not suitable for predicting entities in future timestamps, thus prompting research into extrapolation reasoning. The objective of extrapolation reasoning is to forecast facts in future timestamps. Know-Evolve by *Trivedi et al. (2017)* models events using time point processes and predicts future occurrences

based on the conditional probabilities. RE-NET (*Jin et al., 2019*) uses neighborhood aggregators and recurrent event encoders to encode historical facts. RE-NET uses graph structure information but this model only considers local structures. xERTE (*Han et al., 2020*) establishes an interpretable prediction model based on subgraph search. CyGNet (*Zhu et al., 2021*) applies repeated facts in historical sets to predict high-frequency entities. TANGO (*Han et al., 2021*) models the temporal knowledge graph using neural ordinary differential equations with continuous-time reasoning capabilities. TIme Traveler (TITer) by *Sun et al. (2021)* utilizes reinforcement learning techniques to predict target entities based on path search. Similar to TITer, Clue Searching and Temporal Reasoning (CluSTeR) (*Li et al., 2021a*) also uses reinforcement learning to search for possible target entities. RE-GCN (*Li et al., 2021b*) and its extended CEN (*Li et al., 2022a*) model the evolution of entities and relationships at each timestamp to obtain local historical dependencies, and they introduce static attributes to improve prediction results. However, they do not fully utilize long-term information. EvoExplor (*Zhang et al., 2022a*) achieves entity prediction by capturing complex evolutionary theories and community structures of historical facts. Graph Hawkes Transformer (GHT) (*Sun et al., 2022*) proposes a Transformer based point-to-point process model for capturing structural and temporal information. TLogic (*Liu et al., 2022*) and TLmod (*Bai et al., 2023*) use temporal logic regulations collected from temporal knowledge graphs to predict entities. HiSMatch (*Li et al., 2022b*) incorporated two structural encoders to compute representations of query-related history and candidate-related history, and integrates background knowledge into entity representations. TEemporal logiCal grapH networkS (TECHS) (*Lin et al., 2023*) introduces a temporal graph encoder and a logical decoder for TKG reasoning, and puts forward a forward message-passing mechanism. The Pre-trained Language Model with Prompts for temporal Knowledge graph completion (PPT) (*Xu et al., 2023a*) transforms the quadruple into input for the pre-trained model and converts the time intervals between different timestamps into prompts to constituting coherent sentences with semantic information. *Lee et al. (2023)* proposes a model to use in-context learning with large language models for TKG reasoning. The proposed model transformed relevant historical facts into prompts and earned prediction results through token probabilities. GenTKG (*Liao et al., 2024*) capitalizes on a retrieval tactic on the ground of temporal logic rules and valid fine-tuning with few-shot parameters for TKG reasoning. TiRGN (*Li, Sun & Zhao, 2022*) considers the sequential, reduplicative and periodic patterns of historical facts. TiRGN fully recognizes local and global history information. But TiRGN ignores the interference of repeated history on entity prediction. CENET (*Xu et al., 2023b*) utilizes the frequency of historical events and contrastive learning to obtain correlations between historical and unhistorical events in order to predict matching entities. However CENET does not consider the evolution of historical facts, resulting in poor performance.

## Contrastive learning

Contrastive learning has been broadly utilized across several territories, such as computer vision (*Feng et al., 2023*; *Lin et al., 2023*; *Wu et al., 2023b*), muti-modal learning (*Xie et al., 2023*), audio processing (*Hu et al., 2023*) and natural language processing

(*Rethmeier & Augenstein, 2023*; *Zhao et al., 2023*). Recently, contrastive learning has been applied in static knowledge graph reasoning. SimKGC model (*Wang et al., 2022*) introduced three types of negative samples for improving the efficiency of contrastive learning. SimRE (*Zhang et al., 2024*) employed comparative learning to mimic the head and body of rules, and incorporates rule features into the model through simple addition. However contrastive learning has been less applied in TKG reasoning. The CENET model is an example of using contrastive learning for TKG reasoning. This model utilizes contrastive learning to recognize historical and non-historical facts, reducing the interference of historical facts on entity prediction.

## PRELIMINARIES

### Relational graph convolutional network

To adapt GCNs for relational data, the relational graph convolutional network (R-GCN) was developed. R-GCN effectively aggregate local neighborhood information with effect. It is a model that can be used for message passing.

R-GCN was initially applied to static knowledge graph reasoning to predict missing information in the knowledge graph. The node features of each layer in R-GCN are calculated from the node features of the previous layer and the relationships between nodes, and so as to hold on to the information of the nodes themselves, R-GCN also incorporates self-connection.

$$h_i^{(l+1)} = \sigma \left( \sum_{r \epsilon \mathcal{R}} \sum_{j \in N_i^r} \frac{1}{c_{i,r}} W_r^{(l)} h_j^{(l)} + W_0^{(l)} h_i^{(l)} \right) \tag{1}$$

where $N_i^r$ denotes neighboring nodes with r-type edges connected to node i. $c_{i,r}$ is a constant and it is broadly used to represent the in degree of node i. $W_r^{(l)}$ and $W_0^{(l)}$ are learnable parameters. l denotes the number of layers. $h_i^{(l)}$ denotes the embedding representation of node i in layer l. $W_0^{(l)} h_i^{(l)}$ denotes self-connection.

### Contrastive learning

The learning framework of contrastive learning is based on discriminative representation of contrastive thinking, primarily used for unsupervised or self-supervised representation learning. The notion of contrastive learning is to compare a given sample with similar positive samples and dissimilar negative samples (*Chen et al., 2020*). The use of contrastive learning model and loss function, representations corresponding to positive samples are closer in the representation space, whereas representations corresponding to negative samples are further away in the representation space. In contrastive learning, a small batch of randomly selected N samples is used to obtain enhanced samples. Given a pair of positive samples (i, j), optimize the following loss function using the original and augmented samples.

$$\mathcal{L}_{i,j} = -log \frac{exp(z_i \cdot z_j / \tau)}{\sum_{k=1, k \neq i}^{2N} exp(z_i \cdot z_k / \tau)} \tag{2}$$

where 2N is the sum of the number of the original and augmented samples. $z_i$ is the projection embedding of the sample i and $\tau$ is a temperature parameter facilitating the model learn from hard negatives, and · is dot product that is utilized for calculating the similarity of samples between different views.

## Temporal knowledge graph preliminaries

A TKG $\mathcal{G}$ is formalized as a sequence of KG snapshots, i.e., $\mathcal{G} = \{\mathcal{G}_1, \mathcal{G}_2, ..., \mathcal{G}_T\}$. Each snapshot $\mathcal{G}_t = (\mathcal{E},\ \mathcal{R},\ \mathcal{F}_t)$ can be seen as a directed multi-relational graph at timestamp t. A fact in $\mathcal{F}_t$ is denoted as a quadruple (s, r, o, t), where s, o $\epsilon$ $\mathcal{E}$ and r $\epsilon$ $\mathcal{R}$. It indicates that there is a relationship r between s and o at time t.

The intention of TKG extrapolation is to forecast missing subject entity or object entity in given query (s, r, ?, t) or (?, r, o, t) according to previous historical KG snapshot $\{\mathcal{G}_1, \mathcal{G}_2, ..., \mathcal{G}_{t-1}\}$. For simplicity, this article adds the quadruple of the inverse relation (o, $r^{-1}$, s, t) in TKG. Therefore the extrapolation task is simplified to predict the object entity. For each prediction task (s, r, ?, t) at time t, We employ the TKG snapshot sequence of m timestamps before time t as $\mathcal{G}_{t-m:t-1}$.

# THE PROPOSED METHODOLOGY

The prevailing focus in contemporary research on temporal knowledge graph reasoning lies in extrapolation knowledge graph reasoning, aimed at forecasting future events in the ground of historical occurrences. The proposed TRCL model addresses the task of extrapolation relationship reasoning. To facilitate comprehension, this work introduces the key notations, followed by a systematic presentation of each component of the model. The flow chart of the proposed TRCL framework is displayed in Fig. 1.

## Model overview of the proposed TRLC model

The TRLC model proposed in this article consists of four modules. The first module takes the sequential information of k snapshots before the current timestamp t as input to the recurrent encoder, obtaining the embedding of entities and relations. The second module is to establish a global historical matrix, with the aim of integrating all repeated historical information into knowledge graph reasoning. The third module is to use the loss function of contrastive learning to diminish the effect of irrelevant historical information on the model's prediction of future events. The fourth module uses a time decoder to capture periodic information, and ultimately this module can obtain the predicted results for each entity. The entire framework of the proposed TRLC model is shown in Fig. 2.

## Recurrent encoder

Many models, such as RE-GCN (*Li et al., 2021b*) CEN (*Li et al., 2022a*), and TiRGN (*Li, Sun & Zhao, 2022*), have certified the excellent performance of the R-GCN model in obtaining representations of entities and relations. Unlike RE-GCN, which simply adds representations of entities and relations, TiRGN uses one-dimensional convolution to better combine the representations of entities and relations. We apply the one-dimensional

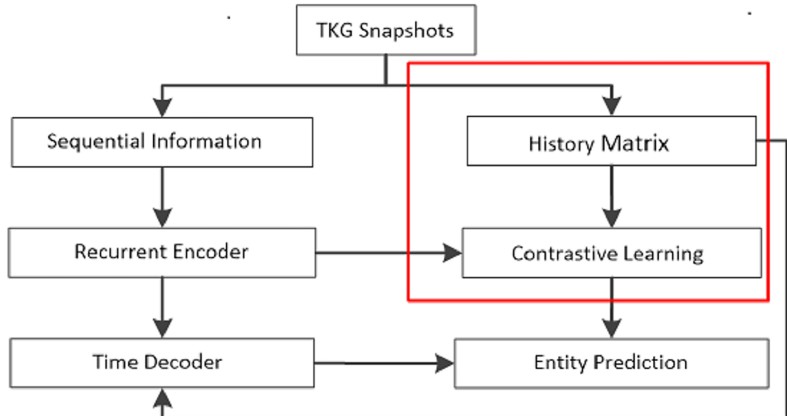

**Figure 1 The flow chart of the proposed TRCL framework.** The red box is the major contribution of this work.

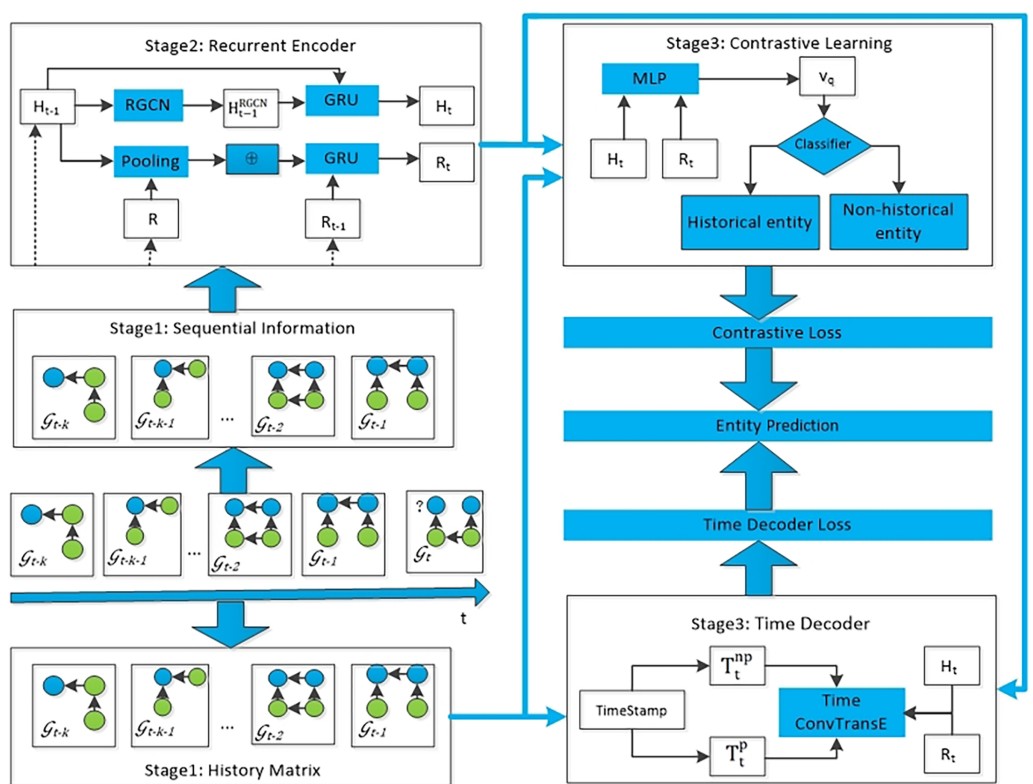

**Figure 2 The entire architecture of TRCL, consists of four modules: recurrent encoder module, history matrix, contrastive learning module and time decoder module.**

convolutional R-GCN introduced by TiRGN to encode entities and relationships at each timestamp. The specific aggregation formula is as follows:

$$h_{o,t}^{l+1} = f\left(\frac{1}{c_0} \sum_{(s,r,o\in\mathcal{F}_t)} w_1^l\left(\varphi\left(h_{s,t}^l, r_t\right)\right) + w_2^l h_{o,t}^l\right) \tag{3}$$

where $h_{s,t}^l$, $h_{o,t}^l$ and $r_t$ denote the embedding representation of entities s, o, and relation r in the $l^{\text{th}}$ layer at time t. $w_1^l$ and $w_2^l$ are learnable parameters for aggregating features and self-loop in the $l^{\text{th}}$ layer. $f(\cdot)$ means the Relu activation function. $\varphi(\cdot)$ denotes the one-dimensional convolution operator. $c_0$ represents the in-degree of entity o, it is a normalizing constant. It is worth noting that when an entity has no relation with other entities in a snapshot, a self-loop edge is still needed to update the representation of that entity.

In order to insure that each query can contain the sequential dependencies of snapshots at the previous timestamp, this article employs the gated recurrent unit (GRU) to gradually renew the representations of entities and relations for each query, ultimately obtaining entity and relation representations that incorporate temporal information from different snapshots. The formula for updating the embeddings of entities in sequence of the snapshots is as follows:

$$H_t = \text{GRU}\left(H_{t-1}, H_{t-1}^{\text{RGCN}}\right) \tag{4}$$

where $H_t$, $H_{t-1} \in \mathbb{R}^{|\varepsilon| \times d}$ are the d-dimensional entity embedding matrices at timestamp t and t − 1, $H_{t-1}^{RGCN} \in \mathbb{R}^{|\varepsilon| \times d}$ is the entity embedding matrix after encoding on the KG snapshots at t − 1. $H_t$ is obtained by calculating $H_{t-1}$ and $H_{t-1}^{\text{RGCN}}$ using the GRU model.

Resemble the formula for updating the embeddings of entities, the formula for updating the embeddings of relations in sequence of the snapshots is as follows:

$$r_t' = \left[\text{pooling}\left(H_{t-1}, V_{r,t}\right); r\right] \tag{5}$$

$$R_t = GRU\left(R_{t-1}, R_t'\right) \tag{6}$$

where $V_{r,t}$ represents the embedding matrices of entities correlated with relationship r at t. $R_t'$ is composed of all relations $r_t'$. $R_t$, $R_{t-1} \in \mathbb{R}^{|\mathcal{R}| \times d}$ denote the embedding matrices of relations at timestamp t and t − 1. $R_t$ is gained by calculating $R_{t-1}$ and $R_t'$ using the GRU model.

## Historical matrix

Considering that certain historical events occur repeatedly, this article establish a historical matrix of entity sets to record whether the current query has appeared in history, in order to provide historical constraints for contrastive learning and time decoder. Specifically, for each query (s, r, ?, t), traverse all snapshots $\mathcal{G}_{0:t-1}$ before t to obtain the corresponding historical entity matrix $M_{s,r,t} \in \mathbb{R}^{|\mathcal{E}| \times |\mathcal{R}| \times |\mathcal{E}|}$. If query (s, r, ?, t) has appeared before t, the value of the corresponding position in the matrix is 1, or else it is 0. We disregard the frequency of quadruples occurrence here, focusing solely on the presence or absence of historical events. This is because events that have repeated numerous times in the past may not necessarily recure in the future. For example, Michael Jordan led the Chicago Bulls to win 6 championships, and when Jordan withdrew from the Chicago Bulls, their chances of winning the NBA championship decreased.

## Contrastive learning

The CENET model as proposed in *Xu et al. (2023b)* uses statistical frequency of historical events and contrastive learning to achieve knowledge graph reasoning. Through

contrastive learning, it distinguishes whether the entity to be predicted by the query is a historical entity or a unhistorical entity. This work draws inspiration from the CENET model. Firstly, there are two types of queries. The first type is the entity to be predicted by the query is a historical entity while the second type is the query is predicting non-historical entities. By minimizing the loss function of contrastive learning to distinguish different types of queries, the same type of query representation is more similar, while different types of query representations are more different. Therefore, minimizing the loss function of contrastive learning can avoid interfering with the prediction results of queries due to irrelevant historical events. Note that the proposed TRCL model differs from the CENET model whereby the module only considers the representation of entities and relations, without considering the frequency of historical events.

The perspective in distinguishing queries is whether the lost entities belong to historical or non-historical entities for each query. This work uses contrastive learning to separate the representations of each query as much as possible. Assume that the embedding representation of q is given:

$$v_q = MLP(h \oplus r). \tag{7}$$

The queried information is encoded by MLP, where h denotes the embedding of entities and r denotes the embedding of relations. The size of the training batches used in this article is divided by timestamps, with each timestamp corresponding to quadruples being a batch. The embedding of entities and relationships stems from the timestamp t corresponding to the current query, so the embedding representation of the query does not need to include time information.

In addition, let $I_q$ represent whether the missing object in query q has appeared in history. If it has appeared, $I_q$ is 1, otherwise $I_q$ is 0. In detail, assuming the predicted result of query q is o, if o has appeared before timestamp t, then $I_q = 1$; otherwise, $I_q = 0$.

$$Q(q) = \bigcup_{m \epsilon M \backslash q} \{m | I_m = I_q\} \tag{8}$$

At last, the loss function for contrastive learning is as follows:

$$\mathcal{L}_{cl} = \sum_{q \in M} \frac{-1}{|Q(q)|} \sum_{k \in Q(q)} \log \frac{exp(v_q \cdot v_k / \tau)}{\sum_{a \epsilon M \backslash \{q\}} (v_q \cdot v_a / \tau)} \tag{9}$$

where M denotes the size of the batch. $\tau$ denotes the temperature parameter. Referring to the CENET model, we set it to 1. Q (q) represents the query m corresponding to the value of $I_q$ in the set M, except for query q. The aim of $\mathcal{L}_{cl}$ is to make the query representations of the same variety closer.

## Time decoder
### The periodic and non-periodic events
Considering that some events occur periodically, for example, the NBA game is held once a year; And some events are non-periodic, such as a certain player will not participate in NBA games after retiring. Therefore, this article take into account the periodicity and

aperiodicity of historical facts. The periodic and aperiodic time-dependent vectors are designed according to Eqs. (9) and (10) by utilizing Time2Vec (*Kazemi et al., 2019*) encoder.

$$T_t^p = sin(\omega_p t + \varphi_p) \tag{10}$$

$$T_t^{np} = \omega_{np} t + \varphi_{np} \tag{11}$$

where $T_t^p$ and $T_t^{np}$ are d-dimensional periodic and aperiodic time vectors, $\omega_p$, $\varphi_p$, $\omega_{np}$ and $\varphi_{np}$ are learnable parameters.

### Time decoder

After obtaining the representation of entity $h_{s,t}$ and relationship $r_t$, as well as the periodic representation of time $T_t^p$ and the non periodic representation of time $T_t^{np}$, we use the convolution operation introduced in TiRGN to fuse the four representations mentioned above. The specific formula is as follows:

$$m_c^n = \sum_{\tau=0}^{K-1} w_c(\tau,0)\hat{h}_{s,t}(n+\tau) + w_c(\tau,0)\hat{r}_t(n+\tau) + w_c(\tau,0)\hat{T}_t^p(n+\tau) + w_c(t,0)\hat{T}_t^{np}(n+\tau) \tag{12}$$

$$M_c = \{m_c^i | \ i \in [0, d-1]\} \tag{13}$$

where c represents the amount of convolutional kernels, n denotes the entries in the output vector ranging from 0 to $d-1$. K denotes the kernel width. $w_c$ is learnable kernel parameters. In addition, padding is applied $h_{s,t}$, $r_t$, $T_t^p$ and $T_t^{np}$ to get $\hat{h}_{s,t}$, $\hat{r}_t$, $\hat{T}_t^p$, $\hat{T}_t^{np}$, respectively. Each convolution kernel can be represented by a vector $M_c$, which can further aligned to obtain the matrix $O_t$.

Conv-TransE (*Shang et al., 2019*) engineers a extraordinary convolution that deformalizes ConvE without compromising prediction performance. Conv-TransE also sustains the translational properties of entities and relationships. Therefore, this work adopts Time ConvTransE as a time dependent decoder to calculate the scores of events, the formula is as described below (*Li, Sun & Zhao, 2022*; *Shang et al., 2019*):

$$\psi(h_{s,t}, r_t, T_t^p, T_t^p) = ReLu(map(O_t)W)H_t \tag{14}$$

$$p^h = softmax(\psi(h_{s,t}, r_t, T_t^p, T_t^p)) \tag{15}$$

where map denotes a feature map operator, and $W \in \mathbb{R}^{cd \times d}$ denotes a matrix for linear transformation. $H_t$ is the embedding of entity.

### Training objective

Because different queries can be duplicate facts or newly occurring facts, we have set a hyper-parameter to equilibrate the values of $p^h$ and $p^{nh}$. The formula is as described below:

$$p = \alpha \times p^h + (1-\alpha) \times p^{nh} \tag{16}$$

where hyper-parameters $\alpha \in [0, 1]$.

The loss function $\mathcal{L}_{td}$ based on time decoder for entity prediction is formalized as follows: where $p(o|s, r, t)$ is the entity prediction probability calculated by Eq. (16), $y_t^e$ is the label vector, which is 1 if the corresponding fact exists, or else it is 0.

The ultimate loss function $\mathcal{L}$ is as described below:

$$\mathcal{L} = \mathcal{L}_{td} + \mathcal{L}_{cl} \tag{17}$$

Note that the loss function $\mathcal{L}_{cl}$ of contrastive learning and the loss function $\mathcal{L}_{td}$ of entity prediction are trained simultaneously.

# RESULTS AND DISCUSSIONS

## Experimental setup

This article uses four datasets to appraise the TRCL models on entity forecast task, including ICEWS14, ICEWS18, ICEWS05-15 (*Lautenschlager, Shellman & Ward, 2015*) and GDELT (*Leetaru & Schrodt, 2013*). These four datasets are widely used for TKG extrapolation. ICEWS14, ICEWS18, and ICEWS05-15 are three subsets of Integrated Crisis Early Warning System (ICEWS). ICEWS contains a large number of political events with timestamps. GDELT is a sub-class of the Global Database of Events, Language datasets, which is also an event set containing temporal information.

Similar to the partitioning methods used in literature REGCN (*Li et al., 2021b*) and Re-Net (*Jin et al., 2019*), we divided all datasets into training, validation, and testing sets in a rate of 0.8, 01, and 0.1. The statistical information of the four datasets is shown in Table 1.

This work applied two evaluation metrics that are extensively used in knowledge graph reasoning, namely mean reciprocal rank (MRR) and H@k (k = 1, 3, 10). MRR represents the average reciprocal values of the ranks of the true entity for all queries, and Hits@k denotes the scale of times that the true entity occur in the top k of the ranked candidates.

Several researches (*Han et al., 2021*; *Sun et al., 2021*) points out that the traditional static filtering setting previously used is not appropriate for extrapolation on temporal knowledge graph reasoning, as only simultaneous facts need to be filtered. Therefore, this work presents the results of the experiment using the recently widely used time aware filtering setting, which only separates out quadruples that occur during query time.

For all datasets, the embedding size d is selected as 200. The amount of one-dimensional convolution-based GCN layers is selected as 2 and the dropout rate for each layer is selected as 0.2. The parameters of TRCL are optimized by using adam (*Kingma & Ba, 2014*) during training, and the learning rate is selected as 0.001. The batch size is selected as the amount of quadruples in each timestamp. During training, the optimal local historical KG snapshot sequence lengths for Integrated Crisis Early Warning System (ICEWS) 14, ICEWS05-15, ICEWS18, and Global Database of Events, Language, and Tone (GDELT) are selected as 12, 19, 17 and 10, respectively. During testing, the optimal local historical KG snapshot sequence lengths for ICEWS14, ICEWS05-15, ICEWS18 and GDELT are selected as 13, 21, 20 and 11, respectively. Like RE-GCN and TiRGN, this article added static KG information to datasets ICEWS14, ICEWS05-15 and ICEWS18. For time-guided decoders, the amount of channels is selected as 50 and the kernel size is selected as 4 × 3. This article attempted numerous α values from 0 to 1 and ultimately determined α = 0.3 as the history weight for all the datasets.

**Table 1 Statistics of the TKG datasets.**

| Dataset | Entities | Relations | Training | Validation | Test | Timestamps | Time interval |
|---|---|---|---|---|---|---|---|
| ICEWS14 | 6,869 | 230 | 74,845 | 8,514 | 7,371 | 365 | 24 h |
| ICEWS05-15 | 23,033 | 251 | 368,868 | 46,302 | 46,159 | 4,017 | 24 h |
| ICEWS18 | 10,094 | 256 | 373,018 | 45,995 | 49,545 | 365 | 24 h |
| GDELT | 7,691 | 240 | 1,734,399 | 238,765 | 305,241 | 2,975 | 15 min |

## Results

The proposed TRCL is compared to eight up-to-date TKG reasoning methods, all of which are typical extrapolated temporal knowledge graph reasoning. These methods include RGCRN (*Seo et al., 2018*), RE-NET (*Jin et al., 2019*), CyGNet (*Zhu et al., 2021*), xERTE (*Han et al., 2020*), TITer (*Sun et al., 2021*), RE-GCN (*Li et al., 2021b*), CEN (*Li et al., 2022a*), TiRGN (*Li, Sun & Zhao, 2022*) and CENET (*Xu et al., 2023b*). The experimental results of the TRCL model and nine baselines on four benchmark datasets are shown in Tables 2 and 3.

## Discussions

According to experimental results, it can be concluded that the proposed TRCL model consistently outperforms RGCRN, RE-NET, RE-GCN, CEN, CyGNet, xERTE, and TITer models. Specifically, although the RGCRN, RE-NET, xERTE, RE-GCN, and CEN models consider adjacent timestamps, the RGCN and GRU models with one-dimensional convolution in the TRCL enhances its ability to recognize structural features and historical correlations more effectively. Therefore, the TRCL model outperforms the RGCRN, RE-NET, RE-GCN, and CEN models. Similar to CyGNet, TRCL also considers repeated historical facts. TRCL's superior performance over CyGNet stems from its comprehensive consideration of the dependency relationships among historical facts and temporal periodicity of facts. Furthermore, the TRCL model outperforms the xERTE and TITer models due to their reliance on subgraph based search and path based search to predict target entities respectively. These methods often face limitations in utilizing long-term information, restricting their ability to recognize complex temporal dependencies. In contrast, TRCL's approach takes into consideration a more nuanced understanding of temporal relationships, resulting in enhanced predictive accuracy.

On the other hand, when compared with the TiRGN model, except for the H@1 metric in dataset ICEWS05-15, the experimental results of the proposed TRCL model attained the best results, indicating that incorporating contrastive learning is profitable for enhancing the performance of our model. Through in-depth analysis of the experimental results of models TRCL and TiRGN, we found that the experimental effect is the best in ICEWS14, the experimental results in ICEWS18 and GDELT were second in effectiveness, the experimental effect is the worst in ICEWS05-15. Through analyzing the dataset, it was found that the dataset of ICEWS14 is the simplest. When comparing with ICEWS14, although the number of timestamps is the same in ICEWS18, this dataset has more entities

**Table 2 Performance (in percentage) for entity prediction task on ICEWS14 and ICEWS18 with time-aware metrics.** The best performance is highlighted in bold.

| Model | ICEWS14 | | | | ICEWS18 | | | |
|---|---|---|---|---|---|---|---|---|
| | MRR | H@1 | H@3 | H@10 | MRR | H@1 | H@3 | H@10 |
| RGCRN (2018) | 38.48 | 28.52 | 42.85 | 58.10 | 28.02 | 18.62 | 31.59 | 46.44 |
| RE-NET (2020) | 39.86 | 30.11 | 44.04 | 58.21 | 29.78 | 19.73 | 32.55 | 48.46 |
| CyGNet (2021) | 37.65 | 27.43 | 42.63 | 57.90 | 27.12 | 17.21 | 30.97 | 46.85 |
| xERTE (2021) | 40.79 | 32.70 | 45.67 | 57.30 | 29.31 | 21.03 | 33.51 | 46.48 |
| TITer (2021) | 41.73 | 32.74 | 46.46 | 58.44 | 29.98 | 22.05 | 33.46 | 44.83 |
| RE-GCN (2021) | 42.00 | 31.63 | 47.20 | 61.65 | 30.58 | 21.01 | 34.34 | 48.75 |
| CEN (2022) | 42.20 | 32.08 | 47.46 | 61.31 | 31.50 | 21.79 | 35.44 | 50.59 |
| TiRGN (2022) | 44.04 | 33.83 | 48.95 | 63.84 | 33.66 | 23.19 | 37.99 | 54.22 |
| CENET (2023) | 32.42 | 24.56 | 35.41 | 48.13 | 26.40 | 17.68 | 29.37 | 43.79 |
| The proposed TRCL | **45.07** | **34.71** | **50.22** | **65.37** | **33.78** | **23.26** | **38.20** | **54.39** |

**Table 3 Performance (in percentage) for entity prediction task on ICEWS05-15 and GDELT with time-aware metrics.** The best performance is highlighted in bold.

| Model | ICEWS05-15 | | | | GDELT | | | |
|---|---|---|---|---|---|---|---|---|
| | MRR | H@1 | H@3 | H@10 | MRR | H@1 | H@3 | H@10 |
| RGCRN (2018) | 44.56 | 34.16 | 50.06 | 64.51 | 19.37 | 12.24 | 20.57 | 33.32 |
| RE-NET (2020) | 43.67 | 33.55 | 48.83 | 62.72 | 19.55 | 12.38 | 20.80 | 34.00 |
| CyGNet (2021) | 40.42 | 29.44 | 46.06 | 61.60 | 20.22 | 12.35 | 21.66 | 35.82 |
| xERTE (2021) | 46.62 | 37.84 | 52.31 | 63.92 | 19.45 | 11.92 | 20.84 | 34.18 |
| TITer (2021) | 47.60 | 38.29 | 52.74 | 64.86 | 18.19 | 11.52 | 19.20 | 31.00 |
| RE-GCN (2021) | 48.03 | 37.33 | 53.90 | 68.51 | 19.69 | 12.46 | 20.93 | 33.81 |
| CEN (2022) | 46.84 | 36.38 | 52.45 | 67.01 | 20.39 | 12.96 | 21.77 | 34.97 |
| TiRGN (2022) | 50.04 | **39.25** | 56.13 | 70.71 | 21.67 | 13.63 | 23.27 | 37.60 |
| CENET (2023) | 39.10 | 29.02 | 43.81 | 58.43 | 20.23 | 12.69 | 21.70 | 34.92 |
| The proposed TRCL | **50.12** | 39.08 | **56.39** | **70.87** | **21.85** | **13.68** | **23.55** | **38.10** |

and data. The experimental effect of ICEWS18 is also not as good as ICEWS14, indicating that contrastive learning is not effective in dealing with complex datasets. The ICEWS05-15 dataset has the highest number of entities and timestamps, but the amount of data is not the largest. In contrast, GDELT has a relatively large number of timestamps but not many entities, and there is extensive training data in GDELT, which enables the model to receive sufficient training. The experimental results of GDELT are much better than those in ICEWS05-15 dataset, indicating that comparative learning with many experimental data can improve classification performance. Moreover, the proposed TRCL model outperforms the CENET model. Specifically, the CENET model takes into account the frequency of contrastive learning and historical facts, however does not incorporate the

**Table 4 The ablation study results on ICEWS14 and ICEWS18 datasets.**

| Model | ICEWS14s | | ICEWS18 | |
|---|---|---|---|---|
| | MRR | H@3 | MRR | H@3 |
| TRCL | 45.07 | 50.22 | 33.78 | 38.20 |
| TRCL-h | 42.76 | 47.73 | 32.60 | 36.65 |
| TRCL-nh | 36.72 | 40.92 | 26.93 | 30.81 |
| TRCL-cl | 44.38 | 49.82 | 33.41 | 37.84 |

model of factual evolution. Therefore, the proposed TRCL model is much higher than the CENET model.

## Ablation study

To investigate the effects of repetitive historical information, recurrent encoder, and contrastive learning on the model TRCL, this work compared different variants of TRCL according to MRR and H@3 metrics: the variant TRCL-h removes repetitive historical information based on the TRCL model; the variant TRCL-nh removed the current encoder from the TRCL model; the variant TRCL-cl has removed contrastive learning from the TRCL model. We have shown the results of the variant model on datasets ICEWS14 and ICEWS18 in Table 4. The results illustrate that the TRCL model surpasses TRCL-h, TRCL-nh, and TRCL-cl in all metrics, which proves that the model can efficiently improve the capability of entity prediction tasks by integrating repetitive historical information, recurrent encoder, and contrastive learning. Specifically, as shown in Table 4.

From the perspective of MRR indicators, TRCL improved by approximately 2.3% on dataset ICEWS14 and 1.2% on dataset ICEWS18 compared to TRCL-h; From the H@3 indicator, TRCL improved by approximately 2.5% on dataset ICEWS14 and 1.6% on dataset ICEWS18 compared to TRCL-h. This indicates that incorporating repetitive historical information into TRCL can improve entity prediction tasks; From the perspective of MRR indicators, TRCL improved by approximately 8.3% on dataset ICEWS14 and 6.8% on dataset ICEWS18 compared to TRCL-nh; From the H @ 3 metric, TRCL improved by approximately 9.1% on dataset ICEWS14 and 7.4% on dataset ICEWS18 compared to TRCL-nh. This indicates that incorporating the current encoder into the TRCL model can effectively identify sequence information of facts.

Meanwhile, from the perspective of MRR indicators, the proposed TRCL model has improved by approximately 0.7% on dataset ICEWS14 and by approximately 0.4% on dataset ICEWS18 compared to TRCL-cl; From the H@3 metric, TRCL improved by approximately 0.4% on dataset ICEWS14 and approximately 0.4% on dataset ICEWS18 compared to TRCL-cl. This indicates that incorporating contrastive learning into TRCL can avoid the interference of repeated historical information on entity prediction tasks.

## Sensitivity analysis

To investigate the influence of repeated historical facts on entity forecast tasks, this work conducted sensitivity analysis on hyper-parameter $\alpha$ in formula 16 on datasets ICEWS14

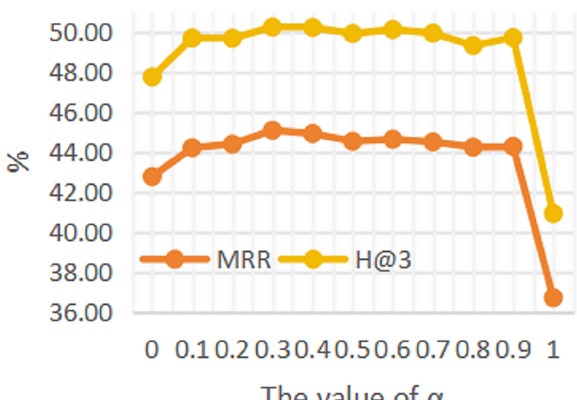

**Figure 3 The sensitivity analysis results of hyper-parameter α on ICEWS14.**

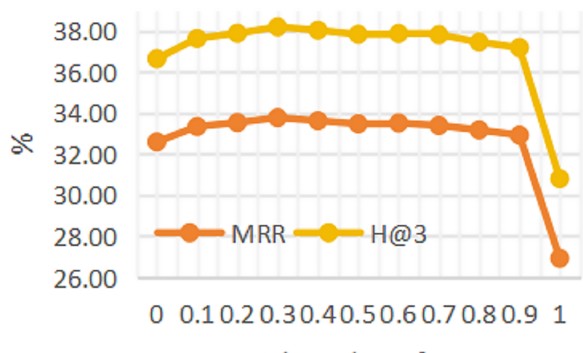

**Figure 4 The sensitivity analysis results of hyper-parameter α on ICEWS18.**

and ICEWS18. The value of a ranges from 0 to 1. The results are shown in Figs. 3 and 4. It can be seen that repeated historical facts have both positive and negative effects on entity prediction tasks. Specifically, on datasets ICEWS14 and ICEWS18, whether it is the MRR metric or the H@3 metric, the performance of entity prediction tasks is optimal when α = 0.3. This indicates that repetitive historical facts contribute to entity prediction tasks, but when this work overly focus on repetitive historical facts, it actually reduces the effectiveness of entity prediction. Our conclusion is also realistic, as historical events that occurred in the past may not necessarily occur in the future.

To investigate the influence of learning rate on entity forecast tasks, this work conducted sensitivity analysis on the learning rate on datasets ICEWS14 and ICEWS18. The values of learning rate are {0.1, 0.01, 0.05, 0.001, 0.005, 0.0001}. The results are shown in Figs. 5 and 6. It can be seen that on datasets ICEWS14 and ICEWS18, the performance

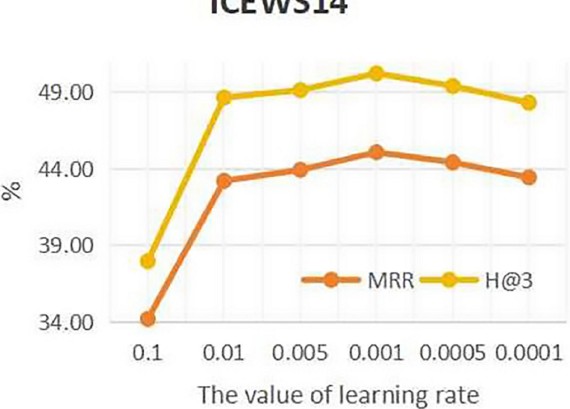

**Figure 5** The sensitivity analysis results of learning rate on ICEWS14.

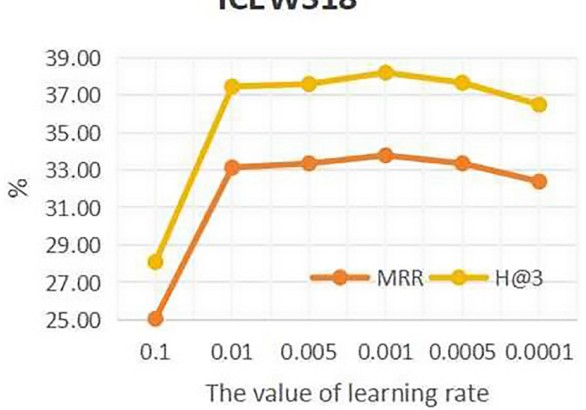

**Figure 6** The sensitivity analysis results of learning rate on ICEWS18.

of entity prediction tasks is optimal when the learning rate is 0.001, for both the MRR metric and the H@3 metric. This indicates that suitable learning rate contributed to good performance in entity prediction tasks.

## CONCLUSION AND FUTURE WORK

This article put forward a new TKG reasoning model, namely TRCL. The TRCL model captures the dependency relationships of historical facts through a recurrent encoder. Afterwards, the model considers the positive impact of repeated historical facts on entity prediction through a global historical matrix. In addition, the model also avoids interference from irrelevant historical facts on entity prediction by incorporating contrastive learning. Finally, the TKG reasoning results are obtained through a time decoder. Substantial experiments conducted on four benchmark datasets have shown that the TRCL model is better than existing methods in most metrics. In the future, this work will study the reasoning ability of models towards emerging facts.

Despite the overall efficacy, TRCL's performance on the ICEWS05-15 dataset highlights the challenge of handling complex data with limited training samples. This requires future enhancements in data efficiency. Future directions include reducing the model's dependence on extensive data, potentially through pre-trained model integration, and broadening its capabilities to relationship prediction tasks within TKG reasoning. These explorations can further solidify TRCL's adaptability and effectiveness in diverse applications, setting a promising path for ongoing research in temporal knowledge graphs.

### Funding
The authors received no funding for this work.

### Competing Interests
The authors declare that they have no competing interests.

### Author Contributions
- Weitong Liu conceived and designed the experiments, performed the experiments, analyzed the data, performed the computation work, prepared figures and/or tables, and approved the final draft.
- Khairunnisa Hasikin conceived and designed the experiments, authored or reviewed drafts of the article, and approved the final draft.
- Anis Salwa Mohd Khairuddin conceived and designed the experiments, authored or reviewed drafts of the article, and approved the final draft.
- Meizhen Liu analyzed the data, prepared figures and/or tables, and approved the final draft.
- Xuechen Zhao performed the experiments, performed the computation work, prepared figures and/or tables, and approved the final draft.

### Data Availability
The Integrated Crisis Early Warning System (ICEWS14, ICEWS18 and ICEWS05-15) datasets are available in the Supplemental File and at Harvard Dataverse: Boschee, Elizabeth; Lautenschlager, Jennifer; O'Brien, Sean; Shellman, Steve; Starz, James; Ward, Michael, 2015, "ICEWS Coded Event Data", https://doi.org/10.7910/DVN/28075, Harvard Dataverse, V37, UNF:6:NOSHB7wyt0SQ8sMg7+w38w== [fileUNF].

The ICEWS14 data was collected from 1/1/2014 to 12/31/2014; the ICEWS18 data was collected from 1/1/2018 to 10/31/2018; the ICEWS05-15 data was collected from 1/1/2005 to 12/31/2015.

The Global Database of Events, Language, and Tone (GDELT) dataset is available in the Supplemental File and at: https://blog.gdeltproject.org/gdelt-2-0-our-global-world-in-realtime/. GDELT is collected from 1/1/2018 to 1/31/2018.

## Supplemental Information

Supplemental information for this article can be found online at http://dx.doi.org/10.7717/peerj-cs.2595#supplemental-information.

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
