# Peer review of "A temporal knowledge graph reasoning model based on recurrent encoding and contrastive learning"

_PeerJ Computer Science, doi:10.7717/peerj-cs.2595_

## Round 0.1 · original submission · Minor Revisions

Some minor revisions are required as per the comments of the reviewers.

·

Basic reporting

Generally, the manuscript is well written however, it should be revised as the following:

1. Quality of figures is so important. Please provide high-resolution figures. Some figures have a poor resolution.
2. The language usage throughout this paper needs to be improved, the author should do some proofreading on it.

Experimental design

1. The paper lacks detailed information on the specific technologies used to work on Knowledge Graphs
2. To enhance the paper´s impact, the author should discuss potential future directions for this research.
3. All images should be in 300 DPI.
4. Try to write mathematical data in a specific mathematical format.
5. The figure label should match the figure.
6. The paper should be enriched with references to existing research papers on knowledge graphs
7. Prepare a Methodology Diagram which should contain the steps and meaningful information.
8. Put a Discussion section separately.
9. Check spelling in Figure 2 (Mentioned by red underline).

Validity of the findings

no comment

Reviewer 2 ·

Basic reporting

The paper is clearly written and provides a comprehensive background on the topic of temporal knowledge graphs (TKGs), focusing on the importance of predicting future events through TKG reasoning. The introduction effectively highlights current research and challenges in this area. The relevant literature is well-referenced, covering various models including RE-NET, CyGNet, and CENET, and comparing with the proposed model TRCL.
As far as I can see, the structure complies with PeerJ standards, and the figures provided are of high quality, well labeled, and support the text. The flowchart in Figure 1 and the architecture diagram in Figure 2 are particularly effective in illustrating the model design.

Experimental design

The paper presents and focused on an important knowledge gap: how to improve the performance of TKG reasoning models in predicting future events. The research question is clearly defined, with the novelty of the TRCL model being its integration of recurrent encoding and contrastive learning to capture dependencies among historical facts while mitigating the interference of repeated historical facts.

The methodology is meticulously designed, and sufficient details are provided for replication. The recurrent encoding mechanism is well described, using a gated recurrent unit (GRU) to incorporate temporal information from historical snapshots, and the historical matrix tracks recurring events. The application of contrastive learning to distinguish relevant and irrelevant historical facts is a strong contribution, addressing a key challenge in TKG reasoning

Validity of the findings

The findings are robust and supported by extensive experiments on four benchmark datasets (ICEWS14, ICEWS18, ICEWS05-15, and GDELT). The results show that TRCL outperforms existing models on most metrics, especially mean reciprocal ranking (MRR) and Hits@k. As highlighted in Tables 2 and 3, the improvements over TiRGN and CENET models are significant, validating the effectiveness of integrating benchmark learning with iterative coding.
The ablation study strengthens the results, showing that each component of the TRCL model contributes to its overall performance. Sensitivity analysis also provides valuable insights into the role of repeated historical facts in forecasting tasks and shows that optimal use of such facts improves the accuracy of the model.

Additional comments

I have two minor suggessions to improve the quality of the paper.
1- As indicated in your results (Lines 373-377), the TRCL model performs relatively poorly on the ICEWS05-15 dataset compared to simpler datasets like ICEWS14. I recommend providing a more in-depth explanation of why TRCL struggles with more complex datasets. It may also be helpful to include suggestions for future improvements or potential modifications to enhance the model’s performance on larger, more intricate datasets
2-The sensitivity analysis in lines 414-423 is insightful, but is primarily focused on the history weight. It would be useful to extend this analysis to include other critical hyperparameters such as learning rate or batch size, as this can help further optimize model performance.

The paper presents a powerful and well-structured approach to solve a difficult problem in TQA reasoning. The innovation of incorporating comparative learning to reduce irrelevant historical facts is particularly commendable. In addition, the rigorous experimental design, including comparisons with several state-of-the-art models and a comprehensive study, strengthens the validity of the findings.

Reviewer 3 ·

Basic reporting

This paper proposes a Temporal Knowledge Graph Reasoning Model based on Recurrent Encoding and Contrastive Learning. Although the technical contribution is somehow incremental, the proposed method is effective and achieves state-of-the-art performance. The authors provide relevant reference for techniques referred from related works.

The problem definition provided in this paper is clear. The authors included most related works of the TKGR task and discussed the contributions and/or limitations of these methods in detail.

The proposed method section clearly elaborates on the four main modules of the proposed method.

This reviewer suggests the authors add a "Preliminaries" section, and move the original "preliminary" subsection and the first paragraph of Section 2.2 (Contrastive Learning) into this section. The related work section should only discuss related works of this paper.

This reviewer recommends that the authors include comparisons with the following related works:
1. Temporal Knowledge Graph Forecasting Without Knowledge Using In-Context Learning (accepted to EMNLP2023 main)

2. GenTKG: Generative Forecasting on Temporal Knowledge Graph with Large Language Models (accepted to NAACL2024 findings).

The GenTKG paper was initially accepted for a poster presentation at the NeurIPS 2023 GLFrontiers Workshop and has been available on ArXiv since 2023. It is important to clarify that this work should not be regarded merely as concurrent with other research in the field.

The enhanced performance of the LLM-based method GenTKG across various evaluation metrics should not diminish the value of traditional embedding and GNN-based approaches. These foundational methods continue to play a critical role in the development of TKGR methods.

Experimental design

The authors conducted experiments on 4 widely-used dataset of the mentioned task, and made comparison with mainstream related works.

The ablation studies demonstrate that each module of the proposed method is crucial for enhancing performance.

This reviewer suggested the authors to compare their proposed method with GenTKG and (Lee et. al 2023) (Please see the Basic Reporting section in my review).

Validity of the findings

No comment.

Additional comments

This reviewer expects a more detailed explanation of the underlying rationales for the model's design choices.
For example, the authors adopt ConvTransE as the time-dependent decoder. This reviewer is interested to see the reasons for selecting this decoder (but not others) in their paper. This could be beneficial for other readers.

---

## Round 0.2 · accepted · Accept

The paper was very well improved!

·

Basic reporting

The authors did the amendment as asked.

Experimental design

Authors completed the the things.

Validity of the findings

Authors completed the the things.

Additional comments

Authors completed the the things.

Reviewer 2 ·

Basic reporting

The paper is clearly written and provides a comprehensive background on the topic of temporal knowledge graphs (TKGs), focusing on the importance of predicting future events through TKG reasoning. The introduction effectively highlights current research and challenges in this area. The relevant literature is well-referenced, covering various models including RE-NET, CyGNet, and CENET, and comparing with the proposed model TRCL.
As far as I can see, the structure complies with PeerJ standards, and the figures provided are of high quality, well labeled, and support the text.

Experimental design

The paper presents and focused on an important knowledge gap: how to improve the performance of TKG reasoning models in predicting future events. The research question is clearly defined, with the novelty of the TRCL model being its integration of recurrent encoding and contrastive learning to capture dependencies among historical facts while mitigating the interference of repeated historical facts.

The methodology is meticulously designed, and sufficient details are provided for replication. The recurrent encoding mechanism is well described, using a gated recurrent unit (GRU) to incorporate temporal information from historical snapshots, and the historical matrix tracks recurring events. The application of contrastive learning to distinguish relevant and irrelevant historical facts is a strong contribution, addressing a key challenge in TKG reasoning

Validity of the findings

The findings are robust and supported by extensive experiments on various benchmark datasets. The results show that TRCL outperforms existing models on most metrics, especially mean reciprocal ranking (MRR) and Hits@k.

The ablation study strengthens the results, showing that each component of the TRCL model contributes to its overall performance. Sensitivity analysis also provides valuable insights.

Additional comments

Thanks to the author for this valuable study. The paper presents a powerful and well-structured approach to solve a difficult problem in TKG reasoning.